# Eating disorders in weight-related therapy (EDIT): Protocol for a systematic review with individual participant data meta-analysis of eating disorder risk in behavioural weight management

Hiba Jebeile[1,2]*, Natalie B. Lister[1,2], Sol Libesman[3], Kylie E. Hunter[3], Caitlin M. McMaster[1], Brittany J. Johnson[4], Louise A. Baur[1,5], Susan J. Paxton[6], Sarah P. Garnett[1,7], Amy L. Ahern[8], Denise E. Wilfley[9], Sarah Maguire[10], Amanda Sainsbury[11], Katharine Steinbeck[1,12], Lisa Askie[3], Caroline Braet[13], Andrew J. Hill[14], Dasha Nicholls[15,16], Rebecca A. Jones[8], Genevieve Dammery[10], Alicia M. Grunseit[5], Kelly Cooper[17], Theodore K. Kyle[18], Faith A. Heeren[19], Fiona Quigley[20], Rachel D. Barnes[21], Melanie K. Bean[22], Kristine Beaulieu[23], Maxine Bonham[24], Kerri N. Boutelle[25], Braulio Henrique Magnani Branco[26], Simona Calugi[27], Michelle I. Cardel[19,28], Kelly Carpenter[29], Hoi Lun Cheng[12], Riccardo Dalle Grave[27], Yngvild S. Danielsen[30], Marcelo Demarzo[31], Aimee Dordevic[24], Dawn M. Eichen[25], Andrea B. Goldschmidt[32], Anja Hilbert[33], Katrijn Houben[34], Mara Lofrano do Prado[35,36], Corby K. Martin[37], Anne McTiernan[38], Janell L. Mensinger[39], Carly Pacanowski[40], Wagner Luiz do Prado[36], Sofia M. Ramalho[41], Hollie A. Raynor[42], Elizabeth Rieger[43], Eric Robinson[44], Vera Salvo[31], Nancy E. Sherwood[45], Sharon A. Simpson[46], Hanna F. Skjakodegard[47], Evelyn Smith[48], Stephanie Partridge[49], Marian Tanofsky-Kraff[50], Rachael W. Taylor[51], Annelies Van Eyck[52,53,54], Krista A. Varady[55], Alaina P. Vidmar[56,57], Victoria Whitelock[58], Jack Yanovski[59], Anna L. Seidler[3], on behalf of the Eating Disorders In weight-related Therapy (EDIT) Collaboration[¶]

**Data Availability Statement:** N/A.

1 The University of Sydney, Children's Hospital Westmead Clinical School, Westmead, New South Wales, Australia, 2 Charles Perkins Centre, The University of Sydney, Sydney, New South Wales, Australia, 3 National Health and Medical Research Council Clinical Trials Centre, The University of Sydney, Sydney, New South Wales, Australia, 4 Caring Futures Institute, College of Nursing and Health Sciences, Flinders University, Adelaide, South Australia, Australia, 5 Weight Management Services, The Children's Hospital at Westmead, Westmead, New South Wales, Australia, 6 School of Psychology and Public Health, La Trobe University, Melbourne, Victoria, Australia, 7 Kids Research, The Children's Hospital at Westmead, Westmead, New South Wales, Australia, 8 MRC Epidemiology Unit, University of Cambridge, Cambridge, United Kingdom, 9 Washington University in St. Louis, St Louis, Missouri, United States of America, 10 InsideOut Institute for Eating Disorders, Charles Perkins Centre, The University of Sydney, Sydney, New South Wales, Australia, 11 The University of Western Australia, School of Human Sciences, Crawley, Western Australia, Australia, 12 The Academic Department of Adolescent Medicine, The Children's Hospital at Westmead, Westmead, New South Wales, Australia, 13 Department of Developmental, Personality and Social Psychology, Ghent University, Henri Dunantlaan, Ghent, Belgium, 14 Leeds Institute of Health Sciences, University of Leeds, Leeds, United Kingdom, 15 Division of Psychiatry, Imperial College London, London, United Kingdom, 16 NIHR ACR Northwest London, London, United Kingdom, 17 Weight Issues Network, New South Wales, Australia, 18 ConscienHealth, Pittsburgh, Pennsylvania, United States of America, 19 Department of Health Outcomes and Biomedical Informatics, University of Florida College of Medicine, Gainesville, Florida, United States of America, 20 Institute of Nursing and Health Research, Ulster University, Newtownabbey, Co. Antrim, Northern Ireland, 21 University of Minnesota Medical School, Minneapolis, Minnesota, United States of America, 22 Department of Pediatrics, Children's Hospital of Richmond at Virginia Commonwealth University, Richmond, Virginia, United States of America, 23 School of Psychology, Faculty of Medicine and Health, University of Leeds, Leeds, United Kingdom, 24 Monash University, Melbourne, Victoria, Australia, 25 Department of Pediatrics, University of California, San Diego, San Diego, California, United States of America, 26 Graduate Program of Health Promotion of University Center of Maringa (UNICESUMAR), Maringa, Parana, Brazil, 27 Department of Eating and Weight Disorders, Villa Garda Hospital, Garda (VR), Italy, 28 WW International, Inc., New York, NY, United States of America,

**Funding:** The EDIT Collaboration is funded by the Australian National Health and Medical Research Council (NHMRC) Ideas Grant (#2002310). Individual author funding: HJ, Sydney Medical School Foundation (University of Sydney), NHMRC Leadership Investigator Grant (#2009035) awarded to LAB; NBL, NHMRC Peter Doherty Early Career Fellowship (GTN1145748); LAB, NHMRC Leadership Investigator Grant (#2009035); ALA and RAJ, Medical Research Council (MRC) (Grant MC_UU_00006/6); AS, NHMRC Senior Research Fellowship (#1135897); DN, National Institute for Health Research (NIHR) under the Applied Health Research (ARC) programme for Northwest London, views expressed in this publication are those of the author(s) and not necessarily those of the National Health Service, the NIHR or the Department of Health in England; DW, Scott Rudolph University Endowed Professorship at Washington University in St. Louis School of Medicine; ALS, NHMRC Emerging Leadership Investigator Grant (#2009432); RDB, National Institute of Health (NIH)/NIDDK (K23 DK092279, R03 DK10400801, R03 DK098492); KC, employed by Optum, the service provider for the weight management program used in the included trial, the funder was not involved in study design, collection, analysis or interpretation of data, writing of the paper and/or decision to submit for publication; DME, NIH (K23DK114480); SMR, doctoral scholarship (SFRH/BD/104182/2014); ER NHMRC (Project Grant #632621; 2010-2012 and Program Grant #1037786); VS, Conselho Nacional de Desenvolvimento Científico e Tecnológico – CNPq – Brazil (Postdoc fellowship); MD, Conselho Nacional de Desenvolvimento Científico e Tecnológico – CNPq – Brazil (research productivity scholarship); ES, receive royalties from Taylor and Francis; AV, eHealth International, Incorporated (AV, http://www.ehealthintl.com/), NIH/NCRR SC-CTSI Grant Number UL1 TR000130, Pediatric Endocrine Society (PES) Clinical Scholar Award (https://pedsendo.org/award/clinical-scholar-awards/), funding agencies are not involved in the design, data collection, analysis, interpretation, or writing. The content is solely the responsibility of the authors and does not necessarily represent the official views of eHealth International, Inc, NIH/NCRR SC-CTSI, PES; VW, Cancer Research UK; SAS, UK MRC and Scottish Chief Scientist Office core funding as part of the MRC/CSO Social and Public Health Sciences Unit 'Complexity in Health Improvement' programme (MC_UU_00022/1 and SPHSU16)]; KAV, NIH (Grant numbers: R01DK128180; R01CA257807; R01 DK119783); MIC, National Institute of Health National Heart, Lung, and Blood Institute K01HL141535; SRP,

**29** Optum Center for Wellbeing Research, Seattle, Washington, United States of America, **30** Department of Clinical Psychology, University of Bergen, Bergen, Norway, **31** Mente Aberta, The Brazilian Center for Mindfulness and Health Promotion, Univesidade Federal de São Paulo, UNIFESP, Brazil, **32** Department of Psychiatry, University of Pittsburgh School of Medicine, Pittsburgh, Philadelphia, United States of America, **33** Research Unit Behavioral Medicine, Integrated Research and Treatment Center Adiposity Diseases, Department of Psychosomatic Medicine and Psychotherapy, University of Leipzig Medical Center, Leipzig, Germany, **34** Department of Clinical Psychological Science, Faculty of Psychology and Neuroscience, Maastricht University, Maastricht, Netherlands, **35** Department of Psychology, California State University, San Bernardino, California, United States of America, **36** Department of Kinesiology, California State University, San Bernardino, California, United States of America, **37** Pennington Biomedical Research Center, Baton Rouge, Louisiana, United States of America, **38** Division of Public Health Sciences, Fred Hutchinson Cancer Center, Seattle, Washington, United States of America, **39** Department of Clinical and School Psychology, Nova Southeastern University, Fort Lauderdale, Florida, United States of America, **40** Department of Behavioral Health and Nutrition, University of Delaware, Newark, Delaware, United States of America, **41** Psychology Research Centre, School of Psychology, University of Minho, Campus Gualtar, Braga, Portugal, **42** Department of Nutrition, University of Tennessee, Knoxville, Tennessee, United States of America, **43** Research School of Psychology, Australian National University, Canberra, Australia, **44** Department of Psychology, University of Liverpool, Liverpool, United Kingdom, **45** Division of Epidemiology and Community Health, School of Public Health, University of Minnesota, Minneapolis, Minnesota, United States of America, **46** Medical Research Council/Chief Scientist Office Social and Public Health Sciences Unit, School of Health and Wellbeing, University of Glasgow, Glasgow, United Kingdom, **47** Department of Clinical Science, University of Bergen, Bergen, Norway, **48** School of Psychology, Western Sydney University, Sydney, New South Wales, Australia, **49** Engagement and Co-design Hub, School of Health Sciences, Faculty of Medicine and Health, The University of Sydney, Sydney, New South Wales, Australia, **50** Departments of Medical and Clinical Psychology and Medicine, Uniformed Services University of the Health Sciences, Bethesda, Maryland, United States of America, **51** Department of Medicine, University of Otago, Dunedin, New Zealand, **52** Laboratory of Experimental Medicine and Pediatrics, University of Antwerp, Antwerp, Belgium, **53** Member of the Infla-Med Centre of Excellence, University of Antwerp, Antwerp, Belgium, **54** Department of Pediatrics, Antwerp University Hospital, Edegem, Belgium, **55** University of Illinois Chicago, Department of Kinesiology and Nutrition, Chicago, Illinois, United States of America, **56** Children's Hospital Los Angeles and Keck School of Medicine of University of Southern California, Los Angeles, CA, United States of America, **57** Department of Pediatrics, Center for Endocrinology, Diabetes and Metabolism, Los Angeles, California, United States of America, **58** Cancer Research UK, London, United Kingdom, **59** Section on Growth and Obesity, *Eunice Kennedy Shriver* National Institute of Child Health and Human Development (NICHD), Division of Intramural Research, National Institutes of Health (NIH), Bethesda, Maryland, United States of America

¶ Members of the EDIT Collaboration are noted in the acknowledgments.
* hiba.jebeile@sydney.edu.au

## Abstract

The Eating Disorders In weight-related Therapy (EDIT) Collaboration brings together data from randomised controlled trials of behavioural weight management interventions to identify individual participant risk factors and intervention strategies that contribute to eating disorder risk. We present a protocol for a systematic review and individual participant data (IPD) meta-analysis which aims to identify participants at risk of developing eating disorders, or related symptoms, during or after weight management interventions conducted in adolescents or adults with overweight or obesity. We systematically searched four databases up to March 2022 and clinical trials registries to May 2022 to identify randomised controlled trials of weight management interventions conducted in adolescents or adults with overweight or obesity that measured eating disorder risk at pre- and post-intervention or follow-up. Authors from eligible trials have been invited to share their deidentified IPD. Two IPD meta-analyses will be conducted. The first IPD meta-analysis aims to examine participant level factors associated with a change in eating disorder scores during and following a weight management intervention. To do this we will examine baseline variables that predict change in

NHMRC and National Heart Foundation Early Career Fellowship (#1157438); MTK, Pilot Intramural Research Award 72ON-01 from USUHS; JAY, National Institutes of Health Intramural Research Program Grant 1ZIAHD000641 from NICHD. JAY reports unrelated grant funds to NICHD supporting his research from Soleno Therapeutics, Rhythm Pharmaceuticals, and Hikma Pharmaceuticals; KNB, NIH grants R01DK116616, R01DK114794, R01 DK111106, R01DK10868, R01DK103554, DOD W81XWH-18-1-0220, R01DK122504, R01 DK075861, HRSA UA3 MC25735, K02HL112042, R21 DK80266 and consults for the Training institute for child and adolescent eating disorders (train2treat4ed.org); JLM, Edna G. Kynett Memorial Foundation Grant (# not applicable) received by JLM https://kynett.org/; AM, Breast Cancer Research Foundation grants (BCRF-16-106/BCRF-17-105/BCRF-18-107/BCRF-19-107/BCRF-20-107/BCRF-21-107), National Cancer Institute/NIH (R01CA105204 (AmcT); U54CA116847 (AMcT); R21 CA131676 (AMcT)) and NIH/NCI Cancer Center Support Grant P30 CA015704. Funders had no role in study design or manuscript preparation for this protocol. Funding for trials which have agreed to join the EDIT Collaboration and share de-identified individual participant data are listed in S3 Table.

**Competing interests:** I have read the journal's policy and the authors of this manuscript have the following competing interests: AS owns 50% of the shares in Zuman International, which receives royalties for books AS has written and payments for presentations. AS additionally reports receiving presentation fees and travel reimbursements from Eli Lilly and Co, the Pharmacy Guild of Australia, Novo Nordisk, the Dietitians Association of Australia, Shoalhaven Family Medical Centres, the Pharmaceutical Society of Australia, and Metagenics, and serving on the Nestlé Health Science Optifast VLCD advisory board from 2016 to 2018. ALA is Principal Investigator on two publicly funded trials where the intervention is provided by WW (formerly Weight Watchers) at no cost. KS has received in kind support as meals from 'Lite and Easy' for a clinical trial of weight stigma in young women in the last 5 years. ER has previously received research funding from Unilever and the American Beverage Association for unrelated work. JAY reports unrelated grant funds to NICHD supporting his research from Soleno Therapeutics, Rhythm Pharmaceuticals, and Hikma Pharmaceuticals. HFS has previously received a salary from Novo Nordisk unrelated to the present work. YSD has previously received a salary from Novo Nordisk unrelated to the present work. HAR has received funding from the National Institutes of

eating disorder risk within intervention arms. The second IPD meta-analysis aims to assess whether there are participant level factors that predict whether participation in an intervention is more or less likely than no intervention to lead to a change in eating disorder risk. To do this, we will examine if there are differences in predictors of eating disorder risk between intervention and no-treatment control arms. The primary outcome will be a standardised mean difference in global eating disorder score from baseline to immediately post-intervention and at 6- and 12- months follow-up. Identifying participant level risk factors predicting eating disorder risk will inform screening and monitoring protocols to allow early identification and intervention for those at risk.

## Introduction

Obesity and eating disorders share some risk factors [1, 2], treatment principles [3, 4], and may co-exist [1, 5–7]. Behavioural weight management is often the cornerstone of first line obesity management [8]. Interventions addressing diet quality, physical activity, and sleep, underpinned by behaviour change strategies, show modest improvements in weight-related outcomes and cardiometabolic health in the short to medium term [9, 10]. However, several common features of behavioural weight management, such as weight loss and dietary restraint are part of eating disorder pathology. Therefore, there is growing international concern among eating disorder and obesity practitioners, professional associations, and advocates that interventions used for weight management may increase the likelihood of participants developing an eating disorder [11, 12].

Systematic reviews show that for most adolescents and adults, medically supervised obesity treatment does not worsen eating disorder risk, at least during the intervention and early follow-up period, and may indeed result in modest improvements [13–15]. Other markers of psychosocial health related to the development of eating disorders, such as depression, anxiety, self-esteem, body image, and quality of life, also show modest improvements after weight management interventions [16–23]. Nevertheless, people with eating disorders are more likely to present for weight loss support than for eating disorder treatment [24] and individual trials have identified a small subset of people who experience the onset of symptoms of eating disorders during, or following, weight management [25–29]. Considering the serious and potentially lifelong consequences of eating disorders [30], it is important that individuals at risk are identified, that treatment options be tailored to ameliorate this risk and that the longer-term outcomes of weight management interventions be adequately assessed.

The Eating Disorders In weight-related Therapy (EDIT) Collaboration will, for the first time, explore the complex risk factor interactions that may precede changes in eating disorder risk during and following weight management (www.editcollaboration.com) [31]. The EDIT Collaboration will address four broad aims: 1) To understand which participants experience a change in eating disorder risk, or related symptoms, during and following weight management interventions; 2) To understand which components of weight management interventions may contribute to change in eating disorder risk; 3) To identify predictive pathways for changes in eating disorder risk during and following weight management interventions; and 4) To develop resources and recommendations to prevent or reduce eating disorder development in response to obesity treatment. The present study will address the first aim of the EDIT Collaboration. No individual trial alone will have a sufficient sample size to address this issue, considering eating disorder risk may emerge in a small subset of individuals undergoing weight

Health in the area of adult and pediatric weight management. HAR is a committee member for the evidence-based practice guidelines for pediatric weight management for the American Psychological Association and for the Evidence Analysis Library for the Academy of Nutrition and Dietetics for the topic of adult weight management and the prevention of type 2 diabetes. MIC is an employee and shareholder at WW International, Inc. TKK has received professional fees from Novo Nordisk, Nutrisystem, Gelesis and Johnson & Johnson. CKM has received research grants and research agreements from Commission on Dietetic Registration, Academy of Nutrition and Dietetics, Ohio State University (InFACT), Novartis, University of Michigan's Michigan Institute for Clinical and Health Research, Elizabeth Blackwell Institute for Health Research, Egg Board, PCORI, Department of Defense, Access Business Group International LLC, IDEA Public Schools, Louisiana LIFT Fund, WW, Pack Health, American Society for Nutrition, RAND Corporation, Richard King Mellon Foundation (RKMF), The Henry M. Jackson Foundation for the Advancement of Military Medicine, Inc., Evidation Health, Leona M. and Harry B. Helmsley Charitable Trust, State of Louisiana- Federal American Rescue Plan (ARP), United States Department of Agriculture (USDA), National Institute for Health Research (NIHR), National Science Foundation (NSF), Lilly, National Institutes of Health (NIH). CKM has served on advisory boards for EHE Health, Wondr Health, and the Nutrition Obesity Research Center at the University of Alabama Birmingham and consulted to Kitchry, Metagenics, WW, Florida Hospital, Gila Therapeutics, Zafgen, OpenFit/MXCXM Health Inc. CKM developed intellectual property (IP) to quantify dietary adherence and his institution has licensed this IP, resulting in receiving royalties via the institution from the licensing fees. CKM is part of US and European patent applications for a weight loss approach called the Body weight Management and activity tracking system and also occasionally gives lectures and talks where he is provided with an honorarium, including talks to the Obesity Action Coalition and Indiana University Bloomington. Finally, CKM serves as a developer and facilitator for continuing education events sponsored by the Commission on Dietetic Registration, and is a Planning Committee Member for the Bray Course. The opinions and assertions expressed herein are those of the authors and are not to be construed as reflecting the views of the Public Health Service, the Department of Health and Human Services, USUHS, or the U.S. Department of Defense.

management and event rates may be low. Thus, novel data analytic approaches that bring together data from multiple trials are needed to determine whether baseline participant characteristics or characteristics of weight management interventions can identify individuals at risk of developing eating disorders.

## Need for individual participant data meta-analysis

Individual participant data (IPD) meta-analysis involves synthesising raw line-by-line participant data across trials. It is considered the "gold standard" for meta-analysis [32], due to its many benefits. For instance, combining IPD from trials enables a larger pooled sample size and greater flexibility to align outcome definitions, thereby increasing statistical power to detect effects for rarer outcomes (such as the emergence of eating disorders) and in smaller samples compared to individual trials alone [33]. It can also lead to improved quantity and quality of available data for analyses by providing access to unpublished outcomes for all randomised participants and by enabling extensive data checking [34]. Further, by leveraging data from existing, previously funded trials, IPD meta-analysis reduces research waste and enables the rapid generation of new knowledge with immediate clinical implications. Of particular importance, collection of data at the individual level provides the opportunity to conduct more in-depth subgroup analyses than using standard aggregate data meta-analyses [34]. Analyses of participants enrolled in various behavioural interventions across diverse subgroups will also enhance generalisability. Overall, this methodological approach will support the identification of individuals at risk of eating disorders in the context of weight management.

## Study aim and objectives

This study aims to understand which participants experience a change in eating disorder risk, or related symptoms, during and after behavioural weight management interventions conducted in adolescents or adults with overweight or obesity. This aim will be addressed by the following objectives: 1) to identify baseline participant risk factors which predict an increase or decrease in eating disorder risk, or related symptoms, during and following a weight management intervention; and 2) to identify whether there are baseline participant risk factors which predict change in eating disorder risk, or related symptoms, if they receive any behavioural weight management intervention compared with no intervention (i.e., no-treatment controls). These two objectives will be addressed in two separate but related studies, outlined in this protocol.

## Methods

We will conduct a systematic review and IPD meta-analysis according to guidelines recommended by the Cochrane Collaboration and Cochrane IPD Meta-analysis Methods Group [34]. This protocol follows the Preferred Reporting Items for Systematic Reviews and Meta-Analysis extension for protocols (PRISMA-P) [35] and was prospectively registered on PROSPERO (CRD42021265340).

## Eligibility criteria

**Types of studies.** This systematic review will include randomised controlled trials with randomisation at the individual level. Quasi- and cluster-randomised and cross-over trials will be excluded. There are no limitations based on date or language.

**Trial participants.** Eligible participants will be adolescents (aged 10 to <19 years at baseline) or adults (aged ≥ 19 years at baseline) with overweight or obesity, as reported by trialists and/or defined as body mass index (BMI) z-score >1 or BMI ≥ 85th percentile in adolescents and BMI ≥25 kg/m$^2$ in adults.

**Types of interventions.** Behavioural weight management interventions aiming to improve a weight-related outcome (e.g., weight/ body mass/ BMI percentile maintenance or reduction) will be included. Interventions may be individual-, group-, or family-based and be conducted in-person or virtually. Trials aimed at preventing obesity in a population classified as overweight will be included. Trials aimed at health promotion or obesity prevention in a broader population including individuals in a healthy weight range will be excluded. To minimise heterogeneity, interventions such as bariatric surgery, post-surgical interventions, pharmacotherapy and nutrient supplementation will be excluded, as will interventions simultaneously targeting secondary or syndromic causes of obesity (e.g., Prader Willi Syndrome), an alternate medical condition (e.g., type 2 diabetes, sleep apnoea) or eating disorders. There are no limitations based on the duration of intervention or setting (e.g., community, inpatient, outpatient).

Within this study, an intervention arm is defined as any delivery of advice or information relating to dietary change, physical activity, sedentary behaviour, sleep health and/or behaviour change and may include print information, online programs or individual- or group-consultations. Interventions may be novel treatments or delivered as part of, in addition to, or separate to, standard clinical care. The intervention period is defined as any period with ongoing contact with the study team. The follow-up period is defined as a period with no contact or intervention provided. Thus, for example, weight maintenance periods providing ongoing support will be considered part of the intervention period. Minimal contact such as newsletters or strategies to keep in touch and assist with the retention of participants will not be considered an intervention. Due to the anticipated high heterogeneity between behavioural weight management interventions, intervention arms will be categorised into broad subgroups, considering the level of prescription (dietary or exercise) within the intervention, dose and type of support provided.

**Types of comparator/control.** Included trials may include a waitlist, no-treatment or weight-neutral control arm, or provide standard care or an alternate intervention as the comparator group. Weight-neutral interventions are defined as those which promote healthy behaviours to improve physical and psychosocial health and well-being, without promoting weight loss or focusing on body weight, shape or size [36]. Within our analyses, control arms will be defined as those providing no treatment, advice or support during the study period. Weight-neutral intervention arms will be considered control arms but will be analysed separately to no-treatment and wait-list controls. Standard clinical care and alternate intervention arms will be considered intervention arms within analyses.

**Types of outcome measures.** Trials must collect at least one measure of eating disorder symptoms or behaviours at baseline and post-intervention or follow-up using a validated self-report questionnaire (e.g., Eating Disorder Examination Questionnaire, Binge Eating Scale) and/or clinical assessment or diagnostic interview (e.g., Eating Disorder Examination). Included outcomes are an eating disorder diagnosis, global eating disorder score, binge eating score, identified engagement in binge eating or frequency of binge eating behaviours, loss of control of eating and/ or bulimic symptoms or compensatory behaviours.

Trials which are planned or ongoing at the time this EDIT Collaboration IPD protocol was finalised in September 2022, will be eligible for inclusion in a nested prospective meta-analysis, provided their results are not yet known [37]. This novel methodology enables integration of prospective evidence into the EDIT IPD. We will aim to harmonise some outcome collection

across these trials if possible (i.e., agree for trials to collect the same core outcomes using the same or similar scales, where feasible).

## Information sources and search strategy

A systematic search of the electronic databases MEDLINE, Embase, PsycINFO and Scopus and clinical trials registries ClinicalTrials.gov and the World Health Organization's International Clinical Trials Registry Platform (WHO ICTRP) Search Portal [38] were conducted from inception to March 2022 and May 2022 respectively. The complete search strategies for all databases and clinical trials registries are listed in S1 Table. Collaborators will be asked to notify us of any additional eligible trials. Authors will also be able to notify the study team of an eligible trial through the study website (https://www.editcollaboration.com/). Trials will be able to continue to join the Collaboration on an ongoing basis.

Records identified through the database searches were imported into Covidence software (Veritas Health Innovation, Melbourne, Australia) to remove duplicates and to screen against the inclusion/exclusion criteria. Two EDIT Collaboration study team members independently screened records by title and abstract and then by full text. Reasons for full text exclusion were recorded. Discrepancies in study eligibility were resolved through discussion with a third member of the study team consulted if necessary.

## Inviting trialists to join the collaboration

A list of eligible studies identified up to August 2022 can be found in S2 Table. The corresponding author/s of eligible studies have been invited to join the EDIT Collaboration via email. If no response was received after two attempts, other authors on the paper or listed on a registration record were emailed and we attempted to contact trialists using our networks. If authors are unable to be contacted after multiple attempts and the required individual participant data are not publicly available, the trial will be excluded, since our analyses are not possible using published summary data alone. Trialists that have agreed to join the Collaboration, as of August 2022, have been shaded in S2 Table.

## Data collection, management and confidentiality

**Data receipt.** Each participating trial will be invited to share de-identified IPD. A complete list of required variables and preferred coding format will be provided to trial investigators; however, data will be accepted in any interpretable format, and recoded by the EDIT team as necessary. Data transfer will occur via secure data transfer platforms or via a secure institutional email using password-protected zip files, depending on trial investigators' preference. All data will be stored in a customised, central and secure database at the National Health and Medical Research Council Clinical Trials Centre, University of Sydney, in accordance with the *University of Sydney Data Management Policy 2014* and will only be accessible to authorised project team members.

**Data processing.** *Data checking*. Data from each trial will be checked in duplicate for outliers or implausible values, missing values and inconsistencies, including cross-checks against published reports, registration records or data collection sheets. The randomisation and study process will be assessed following standard procedures such as inspecting the balance of participant characteristics across groups and the chronological randomisation sequence. Possible errors and inconsistencies will be clarified with the trialist, and the dataset amended if a consensus with each extractor is reached.

*Data recoding*. Data will be re-coded and re-formatted into the preferred format as required, ensuring standardised variable names, measurement scales and units are used (e.g.,

converting body weight in pounds to kilograms). Data will be re-coded by one study team member and checked by a second team member.

*Data transformation and collation.* Once data checking and coding from all trials are finalised, the data will be combined into a single dataset, with a unique identifier for each trial. When required, new variables will be derived from the combined dataset to address the research questions (e.g., calculation of body mass index or categorization of eating disorder scores based on pre-specified cut-points).

## Individual participant variables

The individual participant data variables to be requested from trialists have been informed by expert consultation (Table 1) [39]. An initial list of individual characteristics potentially relevant to eating disorder risk was drafted in consultation with the EDIT Collaboration Scientific and Stakeholder Advisory Panels. These items were then developed into a survey and circulated internationally to eligible trialists and among obesity and eating disorder professional societies and consumer advocacy organisations. Participants were asked to rate the relevance of each individual characteristic to eating disorder risk within the context of weight management and to identify any variables relevant to eating disorder risk not already included [39]. Variables included in the final analyses will be dependent on the availability of IPD, i.e., we will only be able to include variables for which sufficient data are available.

## Defining change in eating disorder risk

There is a paucity of literature to guide the assessment of and change in eating disorder risk in individuals with overweight or obesity. Few studies have validated self-report eating disorder questionnaires against a structured clinical interview in people with overweight or obesity [40]. Thus, it is challenging to identify normative data in this population to ascertain appropriate cut-points indicating someone may be at risk of, or may meet diagnostic criteria for an eating disorder. Similarly, there are no data to indicate clinically relevant changes in eating disorder risk scores on these self-report questionnaires when used longitudinally.

To identify participants at risk of developing an eating disorder in this study, we will assess the change in eating disorder risk in three ways. A standardised mean difference in eating disorder scores between baseline and post-intervention will be used as the primary outcome to determine a statistically significant change in eating disorder risk as a continuous variable. The Reliable Change Index [41] and pre-determined clinical cut-points will be secondary outcomes for assessments with appropriate data and reliability to inform analyses. The Reliable Change Index provides a measure of statistical and clinical significance while considering scale reliability [41, 42]. The index shows how much, and in what direction, an outcome measure has changed at the individual level, and whether that change is both reliable and clinically significant. Table 2 summarises the eating disorder measures used by included trials and the pre-defined cut-points used for each measure to categorize individuals as being high or low risk.

Questionnaires examining global eating disorder risk, for example, the Eating Disorder Examination Questionnaire or the Eating Disorder Inventory, usually include a sub-scale on dietary restraint. While dietary restraint or dieting is considered an eating disorder risk factor in community samples, dietary change is often a core component of behavioural weight management. This presents a challenge with examining eating disorder risk in the context of weight management interventions. A systematic review examining the change in dietary restraint compared to other eating disorder risk factors found that these sub-scales did not align in their direction of effect [49]. Thus, current measures of dietary restraint do not appear

**Table 1. Individual participant variables.**

| Variable | Descriptor/ examples |
|---|---|
| **DEMOGRAPHICS** | |
| Age | Age at enrollment |
| Sex and gender | Female, male<br>Girl/woman, boy/man, non-binary etc. |
| Ethnicity and race | Ethnicity and race of individual, language spoken at home |
| Family structure and environment | Number of adults and children in the household, marital status, parenting style, feeding practices, social or peer support |
| Socio-economic status | Household income, education, employment, Socio-economic index, history of or current food insecurity or deprivation |
| **WEIGHT STATUS** | |
| Anthropometry | Weight, height, BMI (z-score, percentile), BMI category (overweight, obesity, severe obesity), waist circumference (z-score), waist:height ratio |
| **MEDICAL AND FAMILY HISTORY** | |
| Puberty timing | e.g., age at menarche, tanner staging |
| Age at menopause | |
| Medical history of obesity or chronic disease | Age of onset of obesity, history of bariatric surgery, chronic disease e.g., type 2 diabetes, diet-related chronic disease, obstructive sleep apnoea |
| Eating disorder history | History of an eating disorder or eating disorder treatment |
| Family history of obesity or chronic disease | Parental obesity, type 2 diabetes, diet-related chronic disease, family history of bariatric surgery |
| Family history of an eating disorder | Family history of an eating disorder or eating disorder treatment |
| Medication history | General and those known to effect body weight |
| Life events which may impact upon body weight | e.g., pregnancy, smoking cessation |
| Prior weight loss attempts | Number of attempts, weight loss prior to intervention |
| **PSYCHOSOCIAL HEALTH AND EATING BEHAVIOURS** | |
| Mental health comorbidity | Diagnosis or risk score for depression, anxiety, history of trauma, oppositional defiant disorder (ODD), attention deficit hyperactivity disorder (ADHD), obsessive compulsive disorder (OCD), autism spectrum disorder (ASD), borderline personality disorder, addiction/substance abuse, self-harm, suicide attempt/ideation |
| Personality traits | Perfectionism, openness, neuroticism, agreeableness, extraversion, conscientiousness |
| Psychosocial health | Self-esteem, body dissatisfaction, quality of life/ weight-related quality of life, stress |
| Sleep quality | e.g., Pittsburgh Sleep Quality Index |
| Weight stigma | Weight-bias internalisation, weight-based teasing/bullying/stigma/discrimination, weight-talk (peer/family) |
| Eating behaviors | Disinhibition related to eating, impulsivity related to eating, emotional eating, night eating, grazing, external eating/food responsiveness (i.e., eating in response to external cues), secret eating, self-efficacy towards eating |
| Food rules | Narrow range/limited food choices or acceptability (including food rules, limited access to foods, fussy eating) |
| History of dieting | Previous dieting attempts (supervised or self-directed), type of diet, duration of dieting, number of dieting attempts, age of onset of dieting |
| Dietary restraint/ dieting | Dietary restraint sub-scale (e.g., EDE-Q, DEBQ, TFEQ) |

to be appropriate measures of eating disorder risk in the context of weight management, and existing sub-scales may not adequately distinguish between flexible and rigid restraint. Removal of the dietary restraint subscale used within questionnaires will be examined as part of sensitivity analyses.

Table 2. Eating disorder assessments used by trials eligible for inclusion in the EDIT Collaboration and their pre-specified risk cut-point.

| Assessments used by eligible trials | Literature regarding recommended cut-point | Cut-point to be used |
|---|---|---|
| Binge Eating Scale | Cut-point of 17 recommended based on sensitivity of $\geq 0.85$ when used to detect binge eating disorder in adults presenting for bariatric surgery [43] | 17 |
| Eating Disorder Examination/Children's Eating Disorder Examination | Clinician directed interview to determine ED diagnosis and symptoms. No literature identified regarding cut-point in individuals with overweight or obesity (where a diagnosis has not been recorded) | Yes/ no diagnosis |
| Eating Disorder Examination Questionnaire | Suggested cut-points for adults based on sample of female control participants and patients from specialty eating disorder treatment centres aged 16–66 years [44]:<br>BMI 25–30 kg/m$^2$: 3.15<br>BMI > 30 kg/m$^2$: 3.26<br>Comparable to cut-point of 3.1 in adults with BMI > 25 kg/m$^2$ recommended by Mond et al. [45] | Adults:<br>BMI 25–30: 3.15<br>BMI > 30: 3.26 |
| Eating Attitudes Test/Children's Eating Attitudes Test | Cut-point of 10 or 11 suggested for individuals with higher BMI to obtain a balance between sensitivity and specificity [46, 47]. This reduced the false negative rate from 68% at the usual cut-point of 20 down to 32% but gave false positive rate of 35% and overall misclassification rate of 33% [46] | N/A |
| Eating Disorder Inventory | No literature identified regarding cut-point in individuals with higher BMI | N/A |
| Questionnaire for Eating and Weight Patterns/ Questionnaire for Eating and Weight Patterns-Adolescent Version | Cut-point of 2 demonstrated reasonable sensitivity (0.74) in identifying binge eating disorder among adults with obesity [48] | 2 |

Once the IPD dataset has been collated, these will be used in two separate studies addressing the two objectives.

## STUDY 1: Analysis of IPD from each intervention arm

In Study 1, we will assess whether participant level factors predict an increase or decrease in eating disorder risk during and following a weight management intervention (Fig 1A). Study 1 aims to evaluate whether any factors are prognostic of an outcome for participants who had any intervention, it will not evaluate the causal influence of the intervention.

### Eligibility criteria

Study 1 will include weight management intervention arms only. Waitlist, no-treatment control arms and weight-neutral arms will be excluded. Adults and adolescents will be included, but analysed separately, and separate models will be built for each group.

### Primary and secondary outcome

The primary outcome will be a standardised mean difference in global eating disorder score from baseline to immediately post-intervention and at 6- and 12- months follow-up, adjusting for the standardised global eating disorder scores at baseline [50, 51]. A secondary outcome will be a standardised mean binge eating score and frequency immediately post-intervention and at 6- and 12- months follow-up, adjusting for the standardised binge eating scores at baseline [50, 51].

### Predictors: Individual risk factors

Individual participant characteristics in Table 1 will be tested as predictors of eating disorder risk. These risk factors will be adjusted based on available data in individual studies. If risk factors are not available across studies, they may not be analysed. We will critically discuss any lack of evidence on these factors in the publication of our results.

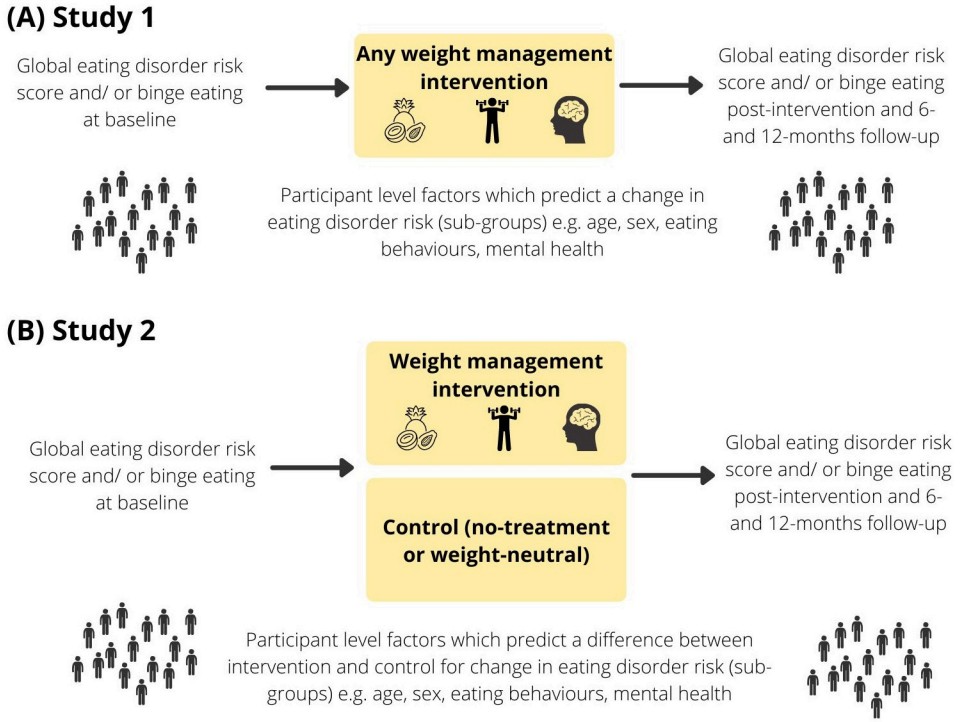

**Fig 1. Individual participant data meta-analyses, (A) Study 1 and (B) Study 2.**

A final set of risk factors to be analysed, will be detailed in the Statistical Analysis Plan. The Statistical Analysis Plan will be finalised and made publicly available on Open Science Framework, with a timestamp, once data collection is finalised but before any analyses are undertaken [34].

## Covariates

Where possible, the impact of the COVID-19 pandemic and related lockdowns will be examined as a covariate in studies collecting data from 2020 onwards.

## Data analysis

**Descriptive analyses.** We will perform the following descriptive analyses:

- Baseline characteristics for each study separately and across studies (sex, age, race, ethnicity, education, socioeconomic status, baseline weight, baseline disordered eating, depression, self-esteem etc.).

- Summary statistics for each risk factor at baseline (for continuous variables, mean and variance; for categorical and binary variables, counts and percentages).

- Eating disorder risk assessments standardised mean and variance before and after interventions (each assessment separately and combined).

- Number of participants with a clinically significant increase or decrease in eating disorder risk using the Reliable Change Index.

- Number of participants moving from below to above, or above to below, a pre-defined cut-point (Table 2) from pre- to post-intervention or from post-intervention to latest follow-up.

Each descriptive analysis will be performed first within each intervention and then across interventions.

## Multilevel models

Separate models will be built for adults and adolescents using the individual participant data, following the same steps. We will build multilevel linear models with random effects for each intervention arm (i.e., each included treatment arm in each study) to account for intervention-level differences. All analyses will be performed using the open-source software R [52], using the package lme4 [53]. Models will be built for the primary outcome (global eating disorder scores) and secondary outcome (binge eating scores). If prognostic variables are highly correlated ($|r| > 0.7$), this may corrupt estimates and model stability, and one of the variables will be removed from the model [54]. The selection of the variable removed will be documented by the authors, and based on its utility, pragmatic qualities and theoretical importance (e.g., which variable is more easily collected, and which variable would be hypothesized to be a more likely causal influence on the outcome). Heterogeneity across interventions will be assessed by calculating the proportion of random variance explained by random intervention effects.

**Model 0: Change of eating disorder risk for all participants.** First, we will build a model to assess if and how eating disorder risk changed for all participants of weight management interventions, only including baseline risk as a predictor, follow-up risk as an outcome, and each intervention as random effects.

**Models 1a,1b,1c.** Then, we will examine univariate models for each participant-level risk factor to assess how this risk factor individually predicts eating disorder risk.

**Model 2.** We will use a backward elimination approach to develop a final model of the key risk factors that influenced eating disorder risk change for all participants. We will start with a global model, then remove the prognostic variable with the least significant value, re-estimate and stop when all *p* values in the multivariate model are significant. The *p* value cut-off chosen will be dependent on the sample size and events per variable (EPV) in our combined dataset (because the number of participants and events can influence statistical significance, lower cut-off values such as 0.05 or 0.01 are recommended for large sample sizes i.e., 100 or more EPV, but a cut off of 0.157 or higher if there are less than 25 EPV) [55, 56]. In the case that the EPV per variable is less than 25 we will consider penalized estimation techniques (LASSO selection) to reduce overfitting. In the case that certain risk factors are systematically missing from a majority of trials because they were not collected, they will be excluded from the global model. Similarly, if trials are excluded from the global model due to only collecting a subset of risk factors relating to a specific research area, we will build additional models using the same process, however now focused on the eating disorder risk within each respective area (e.g., psychological risk factors for ED).

Model fit will be examined using various fit indices (e.g., $R^2$, Cox-Snell generalised $R^2$, Brier Score, calibration plots, residual plots). If we have an indication that there may be a non-linear effect, we will explore the use of cubic splines and compare the linear and non-linear model with a Chi-Square Difference of the Log Likelihood ratio test. Predictive performance will be estimated using various statistics such as C statistics, and calibration slopes applied from the model to each study individually. Predictive performance will then be summarised across studies using a random effect meta-analysis for each statistic. To gauge which predictors have inconsistent effects across studies we will examine the residual intra-class variation [57].

We will present final model parameters including estimated mean difference per standard deviation change for the primary and secondary outcome with 95% confidence interval, standard error and t-value. We will report variance (SD) and explained random variance for random effects.

### Model stability, internal validation, and internal-external cross-validation

Variable selection can often produce optimistic models that have been overfitted to the development dataset and compromise the validity of model predictions for new populations. Due to this possibility, we will probe our model with multiple stability, validation and sensitivity analyses. We will use internal validation to examine overfitting and model stability via the resampling technique of bootstrapping. Bootstrapping will be done within each intervention arm, will be representative of the size of the intervention arm, and will be repeated 100 times. Each resampled dataset allows for the internal validation of how robust the variable selection process is to small deviations in input [55, 56]. Likewise, the development of new model estimates using the new bootstrapped datasets allows for the evaluation of optimism. The performance of the bootstrap models will be evaluated against the bootstrapped *and* original data. Specifically, we will examine the difference in performance of the bootstrap model on both datasets to create an optimism estimate and repeat this process 100 times for each performance statistic. The difference between the initial performance statistic based on the original model and the average of all optimism estimates provides an optimism adjusted performance statistic [58–60]. Internal-external cross-validation will be examined by rerunning the model development process omitting one study, the predictive performance of the model is then examined against the omitted study to externally validate the model. If there is an appropriate sample size [61–63], this process is repeated for each study and meta-analysis, and is used to summarise estimates and heterogeneity to evaluate how reliable the model is when applied to different populations [60]. If any studies have notable case-mix differences (e.g., majority of the sample has high socio-economic status or has class 3 obesity), their omission will also be explored in this cross-validation process to improve the generalisability of the model. We will perform a sensitivity analysis to explore if basic conclusions hold when comparing the final model with those obtained after eliminating some variables from the final model or including additional ones. Finally, we will perform sensitivity analyses to explore model stability with the spread of collinear variables.

## STUDY 2: IPD of randomised controlled trials

In Study 2, we will assess whether there are participant level factors that predict whether participation in any weight management intervention is more or less likely than a no-treatment (e.g., wait-list) control or a weight-neutral intervention to lead to a change in eating disorder risk at post-intervention and 6- and 12-months follow-up (Fig 1B). These two comparisons (comparison of weight management interventions to a no-treatment control and comparison of weight management interventions to a weight-neutral intervention) will be analysed separately to each other. All analyses will be intention-to-treat and include all randomised participants.

### Eligibility criteria

Study 2 will include trials with one or more weight management intervention arms being compared to a waitlist or no-treatment control arm and trials comparing a weight management intervention with a weight-neutral health-focused intervention. Adults and adolescents will be included, but analysed separately, and separate models will be built for each.

### Primary outcome

The primary outcome will be a standardised mean difference in global eating disorder scores between baseline and immediately post-intervention and at 6- and 12- months follow-up.

### Secondary outcome

The secondary outcome will be a standardised mean difference in binge eating scores and frequency between baseline and immediately post-intervention and at 6- and 12- months follow-up.

### Individual risk factors

Individual participant characteristics in Table 1 will be examined as risk factors for eating disorder development. Due to multiple comparisons inflating our type 1 error risk, significant results will be interpreted with caution, assessing the pattern of results instead of single significant findings in isolation, and taking into account the range of uncertainty using the 95% confidence interval [64, 65]. As with Study 1, these risk factors will be adjusted based on the data availability and be detailed further in the Statistical Analysis Plan.

### Data analysis

We will compare individual participant characteristics (Table 1) descriptively at baseline within the intervention and control group for each study individually and across studies. We will describe the number of participants with a statistically significant increase or decrease in global eating disorder or binge eating score in both the intervention and the control group, and the number of participants moving from below to above, or from above to below, a predefined eating disorder risk cut-point (Table 2).

The main analysis will be performed using a random-effect one-stage approach, stratified by trial. To assess whether interventions compared to control affect primary and secondary outcomes overall, we will build multilevel linear models for the continuous primary outcome of global eating disorder score, and other continuous outcomes, and general linear models with a log link function for binary outcomes (e.g., above or below a cut-point). We will be adjusting for eating disorder score at baseline to improve precision, as recommended by Riley et al. [33]. The heterogeneity of treatment effects across trials will also be estimated using $I^2$ which captures the proportion of the total variance driven by heterogeneity, 95% prediction intervals which capture the likely range of effect sizes from an equivalent population of studies, and $\tau^2$ the between study variance. If excessive statistical heterogeneity in treatment effects or inconsistency across trials is detected, then the rationale for combining trials will be questioned and the source of heterogeneity explored.

To analyse individual-level risk factors (i.e., analyse if there are differential treatment effects on eating disorder outcomes for subgroups of participants/ by participant-level predictor variables) differences in intervention effect by risk factor on eating disorder risk score will be analysed by testing an intervention by risk factor interaction term. To account for aggregation bias, this analysis will be stratified by trial, with random effects placed on the within trial interaction [66]. Results will be reported with appropriate estimates, 95% confidence intervals and two-sided p-values of intervention effects and interaction effects.

## Risk of bias and data quality

### Risk of bias assessment

Two reviewers will assess the risk of bias independently with discrepancies resolved through discussion. Each intervention arm in Study 1 will be assessed using the US Academy of

Nutrition and Dietetics Quality Criteria Checklist: Primary Research [67]. The checklist allows a rating (positive, neutral, or negative) of each intervention arm within the study. The Cochrane Risk of Bias Tool [68] will be used to assess risk of bias for trials included in Study 2.

## Data quality and consistency

Data quality will be assessed by examining the IPD and any published materials following a checklist of items derived from the literature and expert consultations, which has been piloted in previous IPDs (*manuscript in progress*). The IPD checks will examine items such as randomisation methods and sequences, inconsistencies in the data or implausible values, retraction notices and ethical approval. Any concerns will be cross-checked with the trialists.

## Publication bias

Potential selection and publication bias will be investigated by comparing IPD meta-analyses of prospectively versus retrospectively included trials in a sensitivity analysis. Where possible, we will include unpublished data, which may alleviate selective outcome reporting bias. Contour-enhanced funnel plots will examine possible differences in results to detect indications of publication bias.

## Confidence in cumulative evidence

Certainty of evidence will be appraised using the Grading of Recommendations Assessment, Development and Evaluation (GRADE) approach [69].

## Missing data

Baseline characteristics of participants completing studies and those who have withdrawn will be compared to identify any differences relevant to eating disorder risk. The principles of Jakobsen et al. [70] will be used to determine whether we will apply complete case analysis (CCA) or multiple imputation. If multiple imputation is the most appropriate approach, we will assume missing at random (MAR) and we will apply a fully conditional (FCS) approach labelled multiple imputation by chained equations (MICE). One regression model will be applied to each trial and treatment group separately [71]. Fifteen imputed dataset copies will be created and meta-analysis will be applied to each dataset and combined using Rubin's rules. Care will be taken to ensure congeniality between the analysis model and the imputation model where possible (e.g., clustering, non-linear effects etc). This process will be applied with the mice package in R [72]. If large proportions (>40%) of data are missing within a trial for an outcome, this trial will be excluded from the primary analysis and included in a sensitivity analysis with complete case analysis to probe whether trials with high missingness influence our estimates.

## Planned sensitivity analyses

Sensitivity analyses will be conducted for Study 1 and 2 excluding interventions with high rates of missing data (>30%), high risk of bias in any domain, trials with a possible conflict of interest and excluding dietary restraint sub-scales within eating disorder scores.

## Data management

Data receipt, management and analysis will be conducted by the National Health and Medical Research Council Clinical Trials Centre, University of Sydney. All data will be stored electronically in a password-protected folder. Storing large de-identified datasets of this kind in

perpetuity is standard procedure for individual participant data meta-analyses, and it is in line with the ICMJE Statement on Data Sharing [73]. Combining data from all trials to-date on weight management interventions measuring eating disorder risk will result in a dataset of unprecedented size for this research question. Storing this data in perpetuity ensures that it can be made available for additional ethics approved purposes that may arise.

## Ethical considerations

IPD will be provided by each trial included in the EDIT Collaboration on the stipulation that the required ethical approval to share data has been obtained by their respective Human Research Ethics Committees (or equivalent), and participants gave informed consent prior to enrolment in the individual trials. Trialists remain the custodians of their data, which will be de-identified before being shared with the EDIT Collaboration. Ethical approval for this project has been granted by The University of Sydney Human Research Ethics Committee (2022/079) and cross-institutional ethics approval from Flinders University Human Research Ethics Committee (project number 5418).

## Additional planned analyses

The following additional planned analyses will be conducted, dependant on additional funding, and a separate protocol will be drafted and made publicly available for each: 1. Change in weight/ BMI related outcomes between baseline and post-intervention and 6- and 12-months follow-up as a predictor of change in eating disorder risk. Where possible for adolescent data, height and weight will be used to standardise BMI across studies (such as using BMI z-score and/ or BMI as a percentage of the 95th percentile); 2. Rate of weight loss between baseline and post-intervention and 6- and 12-months follow-up as a predictor of eating disorder risk; 3. Change in cardiometabolic health outcomes (e.g., lipids, insulin, blood pressure, liver function tests) between baseline and post-intervention and 6- and 12-months follow-up (dependant on additional funding to collate outcomes).; 4. Understanding the temporal change in global eating disorder scores and binge eating scores during weight management interventions and during the follow-up period.

## Discussion

This will be the first study to examine change in individual eating disorder risk in weight management interventions in adolescents and adults with overweight or obesity. Eating disorder risk will be examined in two ways. First, we will identify individual characteristics associated with a change in eating disorder risk following any type of weight management intervention (Study 1). Second, we will examine whether risk varies for participants engaging in a weight management intervention compared to no intervention or a weight-neutral intervention (Study 2). The study will combine data from trials globally to conduct an IPD meta-analysis, which is considered the gold-standard for meta-analysis. Considering that eating disorders are expected to affect a small number of individuals, a greater sample size is required to identify those at risk than is feasible for any trial alone. Findings will inform screening and monitoring protocols to allow individuals at risk to be identified early. Although included data are only available from weight management clinical trials, findings will contribute to our understanding of eating disorder risk in people with overweight or obesity more broadly.

A limitation of this study is the risk of not obtaining data from all eligible trials, leading to inclusion bias. This risk will be minimised by applying a broad set of collaboration building and engagement strategies. Additionally, the wide range of assessments used to measure eating

disorder risk may lead to difficulties with pooling data and harmonising outcomes. We will work closely with the trialists and with IPD experts to address this.

We plan to complete the first round of data collation from included trials by the end of 2022 and conduct the primary analysis by 2023. Trials which have not completed data collection by this time will remain a part of the EDIT Collaboration and will have the opportunity to contribute data to future updates of these analyses.

The EDIT Collaboration [31] will conduct a complementary study that aims to deconstruct weight management interventions into their 'active ingredients' (aim 2 of the EDIT Collaboration), describing delivery features and intervention strategies with potential to increase or decrease eating disorder risk. Upon the completion of these two projects, the resulting data will be used to develop predictive models for the development of eating disorder risk during weight management at the individual level, considering the interaction between individual characteristics and intervention components (aim 3 of the EDIT Collaboration). By improving our understanding of individual experiences of weight management interventions, these models will help inform recommendations for early identification and assessments of eating disorders during weight management and tailored treatment approaches to mitigate future individual risk of eating disorders (aim 4 of the EDIT Collaboration). This IPD meta-analysis is the first step towards understanding individual risk of eating disorder during weight management.

## Supporting information

**S1 Table. Search strategies.**
(PDF)

**S2 Table. Randomised controlled trails eligible for inclusion in the Eating Disorders In weight-related Therapy (EDIT) Collaboration.**
(PDF)

**S3 Table. Funding sources for trials which have agreed to join the EDIT Collaboration.**
(PDF)

## Acknowledgments

Many of the methods and processes described in this protocol have been informed by the TOPCHILD Collaboration methods and procedures [74, 75]. We would like to acknowledge TOPCHILD team members Angie Barba, Jonathan Williams and Mason Aberoumand, who were instrumental in setting up these processes and providing technical support. We would like to thank Mariam Metwally, Timothy Low-wah, Hannah Melville, Eve House and Rabia Khalid for contributions to screening.

## Members of the EDIT Collaboration as at August 2022

**Study team**: Natalie Lister, Hiba Jebeile, Anna L. Seidler, Brittany J. Johnson, Rabia Khalid, Hannah Melville, Sol Libesman, Ruth Tredinnick, Samantha Pryde, Caitlin M. McMaster, Kylie E. Hunter.

**Scientific and Stakeholder Advisory Panel**: Amy L. Ahern, Lisa Askie, Louise A. Baur, Caroline Braet, Kelly Cooper, Genevieve Dammery, Sarah P. Garnett, Rebecca Golley, Alicia Grunseit, Andrew J. Hill, Rebecca A. Jones, Ted K. Kyle, Sarah Maguire, Faith A. Heeren, Dasha Nicholls, Susan J. Paxton, Fiona Quigley, Amanda Sainsbury-Salis, Kate Steinbeck, Denise E. Wilfley, Jacqlyn Yourell.

**Trial representatives**: Rachel Barnes, Melanie Bean, Kristine Beaulieu, Maxine Bonham, Kerri Boutelle, Caroline Braet, Hoi Lun Cheng, Simona Calugi, Michelle Cardel, Kelly Carpenter, Riccardo Dalle Grave, Yngvild Sørebø Danielsen, Marcelo Demarzo, Aimee Dordevic, Dawn Eichen, Andrea Goldschmidt, Anja Hilbert, Katrijn Houben, Natalie B. Lister, Mara Lofrano Prado, Braulio Henrique Magnani Branco, Corby Martin, Anne McTiernan, Janell Mensinger, Faith A. Heeren, Dasha Nicholls, Carly Pacanowski, Stephanie Partridge, Wagner Prado, Sofia Ramalho, Hollie Raynor, Kay Rhee, Elizabeth Rieger, Eric Robinson, Amanda Sainsbury-Salis, Vera Salvo, Brian M Shelley, Nancy Sherwood, Sharon Simpson, Hanna F Skjakodegard, Evelyn Smith, Marian Tanofsky-Kraff, Rachael W. Taylor, Annelies Van Eyck, Krista Varady, Alaina Vidmar, Victoria Whitelock, Denise E. Wilfley, Jack Yanovski.

## Author Contributions

**Conceptualization:** Hiba Jebeile, Natalie B. Lister, Louise A. Baur, Anna L. Seidler.

**Funding acquisition:** Hiba Jebeile, Natalie B. Lister, Louise A. Baur, Susan J. Paxton, Sarah P. Garnett, Amy L. Ahern, Denise E. Wilfley, Sarah Maguire, Amanda Sainsbury, Katharine Steinbeck, Lisa Askie, Anna L. Seidler.

**Methodology:** Hiba Jebeile, Natalie B. Lister, Sol Libesman, Anna L. Seidler.

**Project administration:** Hiba Jebeile.

**Supervision:** Hiba Jebeile, Natalie B. Lister, Susan J. Paxton, Sarah P. Garnett, Anna L. Seidler.

**Writing – original draft:** Hiba Jebeile, Natalie B. Lister, Sol Libesman, Kylie E. Hunter, Anna L. Seidler.

**Writing – review & editing:** Hiba Jebeile, Natalie B. Lister, Sol Libesman, Kylie E. Hunter, Caitlin M. McMaster, Brittany J. Johnson, Louise A. Baur, Susan J. Paxton, Sarah P. Garnett, Amy L. Ahern, Denise E. Wilfley, Sarah Maguire, Amanda Sainsbury, Katharine Steinbeck, Lisa Askie, Caroline Braet, Andrew J. Hill, Dasha Nicholls, Rebecca A. Jones, Genevieve Dammery, Alicia M. Grunseit, Kelly Cooper, Theodore K. Kyle, Faith A. Heeren, Fiona Quigley, Rachel D. Barnes, Melanie K. Bean, Kristine Beaulieu, Maxine Bonham, Kerri N. Boutelle, Braulio Henrique Magnani Branco, Simona Calugi, Michelle I. Cardel, Kelly Carpenter, Hoi Lun Cheng, Riccardo Dalle Grave, Yngvild S. Danielsen, Marcelo Demarzo, Aimee Dordevic, Dawn M. Eichen, Andrea B. Goldschmidt, Anja Hilbert, Katrijn Houben, Mara Lofrano do Prado, Corby K. Martin, Anne McTiernan, Janell L. Mensinger, Carly Pacanowski, Wagner Luiz do Prado, Sofia M. Ramalho, Hollie A. Raynor, Elizabeth Rieger, Eric Robinson, Vera Salvo, Nancy E. Sherwood, Sharon A. Simpson, Hanna F. Skjakodegard, Evelyn Smith, Stephanie Partridge, Marian Tanofsky-Kraff, Rachael W. Taylor, Annelies Van Eyck, Krista A. Varady, Alaina P. Vidmar, Victoria Whitelock, Jack Yanovski, Anna L. Seidler.

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
