## [Decision Letter · Decision Letter 0]

15 Feb 2023

Eating Disorders In weight-related Therapy (EDIT): Protocol for a systematic review with individual participant data meta-analysis of eating disorder risk in behavioural weight management

PONE-D-22-25624

Dear Dr. Jebeile,

We’re pleased to inform you that your manuscript has been judged scientifically suitable for publication and will be formally accepted for publication once it meets all outstanding technical requirements.

Kind regards,

Mohammad Asghari Jafarabadi

Academic Editor

PLOS ONE

1. Please provide additional details regarding participant consent. In the ethics statement in the Methods and online submission information, please ensure that you have specified what type you obtained (for instance, written or verbal, and if verbal, how it was documented and witnessed). If your study included minors, state whether you obtained consent from parents or guardians. If the need for consent was waived by the ethics committee, please include this information.

2.Thank you for stating the following financial disclosure:

“The EDIT Collaboration is funded by the Australian National Health and Medical Research Council (NHMRC) Ideas Grant (#2002310). Individual author funding: HJ, Sydney Medical School Foundation (University of Sydney), NHMRC Leadership Investigator Grant (#2009035) awarded to LAB; NBL, NHMRC Peter Doherty Early Career Fellowship (GTN1145748); LAB, NHMRC Leadership Investigator Grant (#2009035); ALA and RAJ, Medical Research Council (MRC) (Grant MC_UU_00006/6); AS, NHMRC Senior Research Fellowship (#1135897); DN, National Institute for Health Research (NIHR) under the Applied Health Research (ARC) programme for Northwest London, views expressed in this publication are those of the author(s) and not necessarily those of the National Health Service, the NIHR or the Department of Health in England; DW, Scott Rudolph University Endowed Professorship at Washington University in St. Louis School of Medicine; ALS, NHMRC Emerging Leadership Investigator Grant (#2009432); RDB, National Institute of Health (NIH)/NIDDK (K23 DK092279, R03 DK10400801, R03 DK098492); KC, employed by Optum, the service provider for the weight management program used in the included trial, the funder was not involved in study design, collection, analysis or interpretation of data, writing of the paper and/or decision to submit for publication; DME, NIH (K23DK114480); SMR, doctoral scholarship (SFRH/BD/104182/2014); ER NHMRC (Project Grant #632621; 2010-2012 and Program Grant #1037786); VS, Conselho Nacional de Desenvolvimento Científico e Tecnológico – CNPq – Brazil (Postdoc fellowship); MD, Conselho Nacional de Desenvolvimento Científico e Tecnológico – CNPq – Brazil (research productivity scholarship); ES, receive royalties from Taylor and Francis; AV, eHealth International, Incorporated (AV, http://www.ehealthintl.com/), NIH/NCRR SC-CTSI Grant Number UL1 TR000130, Pediatric Endocrine Society (PES) Clinical Scholar Award ( https://pedsendo.org/award/clinical-scholar-awards/), funding agencies are not involved in the design, data collection, analysis, interpretation, or writing. The content is solely the responsibility of the authors and does not necessarily represent the official views of eHealth International, Inc, NIH/NCRR SC-CTSI, PES; VW, Cancer Research UK; SAS, UK MRC and Scottish Chief Scientist Office core funding as part of the MRC/CSO Social and Public Health Sciences Unit ‘Complexity in Health Improvement’ programme (MC_UU_00022/1 and SPHSU16)]; KAV, NIH (Grant numbers: R01DK128180; R01CA257807; R01 DK119783); MIC, National Institute of Health National Heart, Lung, and Blood Institute K01HL141535; SRP, NHMRC and National Heart Foundation Early Career Fellowship (#1157438); KNB, NIH grants R01DK116616, R01DK114794, R01 DK111106, R01DK10868, R01DK103554, DOD W81XWH-18-1-0220, R01DK122504, R01 DK075861, HRSA UA3 MC25735, K02HL112042, R21 DK80266 and consults for the Training institute for child and adolescent eating disorders (train2treat4ed.org); JLM, Edna G. Kynett Memorial Foundation Grant (# not applicable) received by JLM https://kynett.org/; AM, Breast Cancer Research Foundation grants (BCRF-16-106/BCRF-17-105/BCRF-18-107/BCRF-19-107/BCRF-20-107/BCRF-21-107), National Cancer Institute/NIH (R01CA105204 (AmcT); U54CA116847 (AMcT); R21 CA131676 (AMcT)) and NIH/NCI Cancer Center Support Grant P30 CA015704. Funders had no role in study design or manuscript preparation for this protocol. Funding for trials which have agreed to join the EDIT Collaboration and share de-identified individual participant data are listed in Table S3.”

Please respond by return e-mail so that we can amend your financial disclosure and competing interests on your behalf.

“I have read the journal's policy and the authors of this manuscript have the following competing interests: AS owns 50% of the shares in Zuman International, which receives royalties for books AS has written and payments for presentations. AS additionally reports receiving presentation fees and travel reimbursements from Eli Lilly and Co, the Pharmacy Guild of Australia, Novo Nordisk, the Dietitians Association of Australia, Shoalhaven Family Medical Centres, the Pharmaceutical Society of Australia, and Metagenics, and serving on the Nestlé Health Science Optifast VLCD advisory board from 2016 to 2018. ALA is Principal Investigator on two publicly funded trials where the intervention is provided by WW (formerly Weight Watchers) at no cost. KS has received in kind support as meals from ‘Lite and Easy’ for a clinical trial of weight stigma in young women in the last 5 years. ER has previously received research funding from Unilever and the American Beverage Association for unrelated work. JAY reports unrelated grant funds to NICHD supporting his research from Soleno Therapeutics, Rhythm Pharmaceuticals, and Hikma Pharmaceuticals. HFS has previously received a salary from Novo Nordisk unrelated to the present work. YSD has previously received a salary from Novo Nordisk unrelated to the present work. HAR has received funding from the National Institutes of Health in the area of adult and pediatric weight management. HAR is a committee member for the evidence-based practice guidelines for pediatric weight management for the American Psychological Association and for the Evidence Analysis Library for the Academy of Nutrition and Dietetics for the topic of adult weight management and the prevention of type 2 diabetes. MIC is an employee and shareholder at WW International, Inc. TKK has received professional fees from Novo Nordisk, Nutrisystem, Gelesis and Johnson & Johnson. CKM has received research grants and research agreements from Commission on Dietetic Registration, Academy of Nutrition and Dietetics, Ohio State University (InFACT), Novartis, University of Michigan’s Michigan Institute for Clinical and Health Research, Elizabeth Blackwell Institute for Health Research, Egg Board, PCORI, Department of Defense, Access Business Group International LLC, IDEA Public Schools, Louisiana LIFT Fund, WW, Pack Health, American Society for Nutrition, RAND Corporation, Richard King Mellon Foundation (RKMF), The Henry M. Jackson Foundation for the Advancement of Military Medicine, Inc., Evidation Health, Leona M. and Harry B. Helmsley Charitable Trust, State of Louisiana- Federal American Rescue Plan (ARP), United States Department of Agriculture (USDA), National Institute for Health Research (NIHR), National Science Foundation (NSF), Lilly, National Institutes of Health (NIH). CKM has served on advisory boards for EHE Health, Wondr Health, and the Nutrition Obesity Research Center at the University of Alabama Birmingham and consulted to Kitchry, Metagenics, WW, Florida Hospital, Gila Therapeutics, Zafgen, OpenFit/MXCXM Health Inc. CKM developed intellectual property (IP) to quantify dietary adherence and his institution has licensed this IP, resulting in receiving royalties via the institution from the licensing fees. CKM is part of US and European patent applications for a weight loss approach called the Body weight Management and activity tracking system and also occasionally gives lectures and talks where he is provided with an honorarium, including talks to the Obesity Action Coalition and Indiana University Bloomington. Finally, CKM serves as a developer and facilitator for continuing education events sponsored by the Commission on Dietetic Registration, and is a Planning Committee Member for the Bray Course.”

Please respond by return email with your amended Competing Interests Statement and we will change the online submission form on your behalf.

Reviewers' comments:

Reviewer's Responses to Questions

**Comments to the Author**

1. Does the manuscript provide a valid rationale for the proposed study, with clearly identified and justified research questions?

Reviewer #1: Yes

Reviewer #2: Yes

2. Is the protocol technically sound and planned in a manner that will lead to a meaningful outcome and allow testing the stated hypotheses?

Reviewer #1: Yes

Reviewer #2: Yes

3. Is the methodology feasible and described in sufficient detail to allow the work to be replicable?

Reviewer #1: Yes

Reviewer #2: Yes

4. Have the authors described where all data underlying the findings will be made available when the study is complete?

Reviewer #1: Yes

Reviewer #2: Yes

5. Is the manuscript presented in an intelligible fashion and written in standard English?

Reviewer #1: Yes

Reviewer #2: Yes

6. Review Comments to the Author

You may also provide optional suggestions and comments to authors that they might find helpful in planning their study.

Reviewer #1: I would like to congratulate the authors and everyone else who contributed to this manuscript. The paper was well-written following a very comprehensive approach with in-depth detail about the rational and methodology. I believe that this submission is certainly worth publishing as it will contribute to the present body of knowledge. The only thing that I could pick up was that on p.26, row 422, you refer to Table 3. I believe this is a typo and you meant Table 2. Overall, the manuscript is of great standard.

Reviewer #2: This review protocol examines the possibility of developing eating disorders or related symptoms during or after weight control interventions that are advised for obese or overweight patients. Despite this challenging question, the authors proposed the review in a concise, well-structured, and robust methodological framework. Using individual patient data would yield a higher quality of evidence; however, many statistical considerations, as reported in the manuscript, are required to be properly addressed during the conduction of the study.

7. PLOS authors have the option to publish the peer review history of their article (what does this mean?). If published, this will include your full peer review and any attached files.

Reviewer #1: No

Reviewer #2: No

---

## [Editor Report · Acceptance letter]

15 Mar 2023

PONE-D-22-25624 

Eating Disorders In weight-related Therapy (EDIT): Protocol for a systematic review with individual participant data meta-analysis of eating disorder risk in behavioural weight management 

Dear Dr. Jebeile:

I'm pleased to inform you that your manuscript has been deemed suitable for publication in PLOS ONE. Congratulations! Your manuscript is now with our production department. 

Kind regards, 

on behalf of

Professor Mohammad Asghari Jafarabadi 

Academic Editor

PLOS ONE